# Effects of a Personal Health Record in Maternity Care: A Stepped-Wedge Trial

**DOI:** 10.3390/ijerph181910343

**Published:** 2021-09-30

**Authors:** Carola J. M. Groenen, Jan A. M. Kremer, Joanna IntHout, Marjan J. Meinders, Noortje T. L. van Duijnhoven, Frank P. H. A. Vandenbussche

**Affiliations:** Department of Obstetrics and Gynaecology, Radboud University Medical Center, P.O. Box 9101, 6500 HB Nijmegen, The Netherlands; jan.kremer@radboudumc.nl (J.A.M.K.); Joanna.inthout@radboudumc.nl (J.I.); marjan.meinders@radboudumc.nl (M.J.M.); noortje.vanduijnhoven@radboudumc.nl (N.T.L.v.D.); frank.vandenbussche@radboudumc.nl (F.P.H.A.V.)

**Keywords:** maternity care, personal health records, multidisciplinary collaboration, pregnant women

## Abstract

To improve both the active involvement of pregnant women in their maternal health and multidisciplinary collaboration between maternal care professionals, we introduced a personal health record (PHR) in routine maternity care. We studied the effects of this intervention on the percentage of uncomplicated births, women’s perspectives on quality of care, and the collaboration between health care professionals. We performed a stepped-wedge cluster randomized controlled trial with four clusters and 13 maternity health centers (community-based midwife practices and hospitals) in one collaborative area. In total, 7350 pregnant women and 220 health care professionals participated. Uncomplicated births accounted for 51.8% (95% CI 50.1–53.9%) of total births in the control group and 55.0% (CI 53.5–56.5%) of total births in the intervention group (*p* = 0.289). Estimated means revealed that the differences detected in the stepped-wedge study were due to time and not the intervention. Women’s perspectives on quality of care and collaboration between health care professionals revealed no relevant differences between the control and intervention groups. The introduction of the PHR resulted in no significant effect on the chosen measures of quality of maternal care. The suggested positive effect in the raw data was a local trend which was less visible in the national database, and thus might be related to subtle changes toward an improved collaborative culture in the study region.

## 1. Introduction

Personal health records (PHRs) have variable designs and features but are all online applications through which individuals can access, manage, and share their health information in a private, secure, and confidential environment. PHRs are said to empower patients, facilitate communication among health care professionals in the patient’s network, and improve health outcomes [1,2,3,4,5,6,7,8,9]. In maternity care, PHRs show modest but important health effects on women, and promote feelings of control and empowerment [10,11].

Dutch maternity care is divided into primary, secondary, and tertiary care, each of which are managed by different organizations. The vast majority—i.e., 87% of all pregnant women—start their maternity care in a primary care setting. In cases involving risk factors or complications, women are referred to secondary or tertiary care. Approximately half (51%) of pregnant women start childbirth in primary care, and 28% go on to give birth in a primary setting. After birth, 96% of all women receive care at home from a maternity care assistant under the supervision of a community-based midwife [12]. Overall, the Dutch maternity care system involves multiple health care professionals working in different organizations and capacities. Professionals in Dutch maternity care face challenges in the optimization of care for both mothers and children. The active involvement of pregnant women and better collaboration among the relevant health care professionals are two requirements explicitly raised in the new Dutch Guideline for Integrated Maternity Care [13]. Given the documented positive effects of PHRs and the specific request for better multidisciplinary collaboration among all involved maternal care professionals, a PHR for Dutch maternity care, MyPregn@ncy, was developed. Through the use of MyPregn@ncy, professionals of different organizations can maintain collaborative involvement with a pregnant woman. We designed a study to assess the introduction of MyPregn@ncy in one Dutch maternity care region. A process evaluation of the introduction of MyPregn@ncy was published previously [14].

The aim of the present paper is to present the effects of the introduction of MyPregn@ncy on health outcomes in maternity care. The effect of the intervention on quality of care from the women’s perspectives and the effect on collaboration between health care professionals were also studied.

## 2. Materials and Methods

### 2.1. Setting

The study was performed in Nijmegen, a single collaborative regional area in the Netherlands with an average of 4000 births a year and over 220 health care professionals involved in maternity care. In the Netherlands, maternity health care is divided into primary, secondary, and tertiary care [15]. Community-based midwives, maternity care assistants, youth health doctors, and nurses provide primary care. Obstetricians (in training), hospital-based midwives, and pediatricians (in training) provide secondary care in hospitals. Tertiary care takes place in hospitals with an obstetric high care department and a neonatal intensive care unit. In this study, all 13 regional maternity health centers, consisting of eleven community-based midwife practices (primary care) and two hospitals (one secondary and one tertiary care), participated. Therefore, it was possible for all pregnant women in this region to participate in this study. All of the participating regional maternity health centers register their maternity data in the national database Perined. In the Netherlands, more than 95% of all births and the accompanying perinatal outcomes are centrally registered in this large national database.

### 2.2. Intervention

The intervention in this study was the introduction of the PHR by pregnant women in the intervention group and their respective health care professionals. This PHR and other comparable tools were developed by MijnZorgnet.nl and have been used previously for women experiencing infertility and for people with Parkinson’s disease [16,17]. Use of a PHR in a maternity care setting was noteworthy as the introduction of the PHR explicitly meant that the PHR was made available to all pregnant women and professionals.

Pregnant women could register on a secured website to start their online MyPregn@ncy. They decided who was granted access to their MyPregn@ncy and became a member of their personal care team. Therefore, they could invite any health care professionals who they considered to be important to their health and to the care process throughout pregnancy and birth. To ensure safe access to MyPregn@ncy, pregnant women and health care professionals registered and logged in using a personal national identification code.

MyPregn@ncy has several functionalities: 1. communication with one or more health care team members, 2. a diary (blogging feature), 3. a library (storage of important documents), and 4. interactive (medical) modules specifically developed for pregnant women. All team members (the pregnant woman and the professionals to whom she granted access) of one personal MyPregn@ncy could access all fields and could add, act, or react. All activities were logged, so the woman had full insight into all delivered input.

### 2.3. Implementation Strategy

Prior to the introduction of the PHR, maternity health care professionals from all involved maternity centers were trained to explain the use and possibilities of MyPregn@ncy. Each maternity center had one contact-professional for the researchers to communicate through. Professionals were instructed to inform all pregnant women in their health center about MyPregn@ncy, and offer them the opportunity to use it. This provision of information and the offer to use the PHR was considered to be the intervention, independent of the final use of MyPregn@ncy. To achieve optimal propagation of the tool and the accompanying study, we created various information leaflets and launched a supporting website. Furthermore, we visited each maternity health center to explain MyPregn@ncy and the study design.

When a maternal health care center entered the intervention condition (see Study design and cluster randomization, below), all pregnant women that received care in that center were informed about the study and invited by their professional to start their PHR. Both health care professionals and clients signed informed consent for the provision and use of anonymized data on pregnancy and childbirth for the purpose of this specific study. Pregnant women signed the informant consent at their maternity center.

Throughout the study, researchers (CG and ND) were available to answer questions and provide clarification to all users. Finally, involved health care professionals received newsletters on the progress of the study.

### 2.4. Study Design and Cluster Randomization

We performed a stepped-wedge cluster randomized controlled trial to study the effects of the introduction of MyPregn@ncy. We included all primary, secondary, and tertiary maternity health care centers in the area, thereby facilitating multidisciplinary collaboration between health care professionals. In this way, this study constitutes a realistic representation of maternity health care in the Netherlands. In a stepped-wedge trial, all pregnant women, in clusters based on their respective maternity health centers, cross over from a control condition to an interventional condition at different points in time [18,19,20,21]. All health care professionals of the 13 maternity health centers, 11 community-based midwife-practices, and 2 hospitals participated. We formed four clusters (A–D); cluster A crossed over first and cluster D crossed over last. We first randomly allocated the two hospitals to participate in either group A or B due to their important role in cases where a pregnant woman was to be transferred from primary to secondary/tertiary care. Thereafter, the midwife practices were randomly assigned to the clusters (A, B, C, D) such that each eventually contained three or four maternity health centers. Randomization was performed by drawing sealed envelopes containing the names of health centers and assigning them to a cluster.

The time between two successive cross over points was set at three months. Figure 1 illustrates the study design, including a pre-rollout period, four cross over points, and a post-rollout period. The intervention group in this stepped-wedge trial is composed of all pregnant women of the maternity health centers in cluster A in step 1, in cluster A and B in step 2, in clusters A, B, and C in step 3, and in clusters A, B, C, and D in step 4 and the post-rollout period. The control group was composed of all other women. At the start of the study, all pregnant women in maternity health centers were initially in the control condition; at the end of the study, all pregnant women in maternity health centers had switched to the intervention condition.

### 2.5. Outcome Measures and Data Collection

The primary outcome measure was the proportion of uncomplicated births. Births were considered to be uncomplicated when they met all following six conditions: (1) gestational age at time of delivery 37–42 weeks, (2) spontaneous start of labor, (3) vaginal non-instrumental delivery, (4) postpartum hemorrhage < 1000 mL, (5) weight of neonate between 5 and 95%, and (6) APGAR score of >7 after 5 min. These conditions are broadly accepted in the professional maternity care field to distinguish between uncomplicated and complicated births [13,22].

Data were extracted from the Dutch Perinatal Registry [23,24,25], in which more than 95% of all births and accompanying perinatal outcomes are centrally registered by maternity health care professionals. Extraction of data regarding the six abovementioned criteria for uncomplicated labor and birth was performed based on the user codes of the participating maternal health centers and on the dates of birth of the newborns (matching the time frames of our study).

The second endpoint was the effect of the PHR on women’s perspectives of the quality of their care, measured using a simplified version of the validated ReproQ self-administered questionnaire. This instrument was developed to evaluate prenatal, natal, and postnatal care, and is based on the WHO Responsiveness model, which includes eight domains: dignity, autonomy, confidentiality, communication, prompt attention, social consideration, basic amenities, and choice and continuity [26,27,28]. Each domain consisted of several items, through which experiences were rated on a 4-point scale, with ‘1′ being the lowest score and ‘4’ being the highest. While the original questionnaire consisted of antepartum and postpartum questionnaires with a large overlap, we used a single postpartum self-administered questionnaire which contained all items. The Dutch language was used exclusively in this questionnaire.

Each 3-month step of the study was divided into six weeks of preparation and inclusion of the women, and six weeks of data collection. In this way, we collected the data of approximately 50% of the participants at their first post-natal visit at the maternity health center.

The third endpoint was the effect on the collaboration between health care professionals (compared between before and after the intervention). All active health care professionals were invited at the start and again at the end of the study by email to give a score on their perception of the quality of regional collaboration. Scores were between 1 and 10, with 1 being the worst rating and 10 being the best. In most cases, the same health care professional gave her or his scores twice (before and after the study). However, the scores from obstetricians in training were mostly provided by different people due to the rotating nature of their training.

### 2.6. Statistical Analysis

Data analysis was performed according to an intention-to-treat principle. First, individual Perined and ReproQ data were combined with the data from the different maternity health care centers. To estimate the intervention effect on the primary outcome, a generalized linear mixed model with logit link and binomial distribution was applied with uncomplicated birth (yes/no) as the dependent variable, fixed effects for intervention and step, and a random effect for health care center with a variance components covariance structure. Hence, the analysis had some similarities to a time series analysis with multiple time points before and after the intervention [19,29]. The scores pertaining to quality of care from the women’s perspectives were analyzed with descriptive statistics and presented graphically. The collaboration scores, as provided by the professionals, were analyzed with a linear mixed model with the timepoint (before/after the switch) as a fixed effect, and person who filled the questionnaire as a random effect, again with a variance components structure. All data were analyzed using SPSS (version 20.0 for Windows; SPSS Inc., Chicago, IL, USA).

## 3. Results

Throughout the study period (2013–2015), 2890 women were included under the control condition, and 4460 women were included under the intervention condition. Table 1 presents the characteristics of the women from the intervention and control groups and reveals no significant differences between the groups.

Figure 2 presents the uptake of MyPregn@ncy inside the intervention group. Of the initial 4460 women in the intervention group, 88% were offered MyPregn@ncy at their maternity health care center. Only 4% of this group began using MyPregn@ncy and, thereafter, 83% continued using it. There was large variation in participation between maternity health centers. One maternity health center had 0% of pregnant women activate MyPregn@ncy, while 90% of pregnant women from another maternity health center activated it.

Complete data for determination of our primary outcome measure were obtained in >90% of the registered women (Table 2).

The percentage of uncomplicated births was 51.8% (95% confidence interval (CI) 50.1–53.9%) for the control group and 55.0% (CI 53.5–56.5%) for the intervention group (*p* = 0.289).

Figure 3 presents the percentages of uncomplicated births for each participating maternity health center both before and after the introduction of MyPregn@ncy.

Figure 4 presents the total percentage of uncomplicated births for both groups in each step of the study. This percentage increased from 45.4% (CI 42.1–47.9%), during the pre-rollout period, to 59.0% (CI 52.0–65.3%) for the women that were included in step 4, and slightly decreased afterwards. Estimated means revealed that the differences in uncomplicated births in the stepped-wedge design were due to time instead of the intervention (*p* = 0.289).

Figure 5 presents the women’s perspectives on quality of care based on the 8-domain WHO Responsiveness model for the control and intervention groups. All mean scores were between 3.60 and 3.90, and the results showed no relevant differences between the control and intervention groups.

Table 3 presents the quality of the regional collaboration as assessed by the maternity health care professionals. The mean total scores were not different between the start and end of the study; hospital-based midwives reported a significantly higher score for collaboration after the introduction of MyPregn@ncy.

## 4. Discussion

We introduced a PHR in a maternal care region using a stepped-wedge design. In total, 7350 women participated in this study. We detected a low percentage of MyPregn@ncy use by the end of the study, combined with a trend toward more uncomplicated births after the intervention of offering the PHR to pregnant women. This trend could not be attributed to our introduced intervention per se, and could be due to a wider positive change in the number of uncomplicated births over the years of our study period. Women’s perspectives on the quality of care scored highly, with no significant differences between the control and intervention groups. Similarly, the scores of health care professionals regarding their perspectives on collaboration revealed no significant differences.

This study evaluated a complex intervention that requires new processes of care, leading to new roles and attitudes of—and communication between—pregnant women and maternity care professionals. Due to the infrastructure involved in the process of offering this innovative PHR, the majority of pregnant women were able to use MyPregn@ncy, if they so wished. However, handling and acceptance of the aforementioned changes by professionals requires them to embrace a change in norms. Our process analyses [14] describe a difference in client and professional expectations of MyPregn@ncy. Therefore, we recommend that discussing the expectations and wishes of pregnant women and professionals should be part of the implementation process. We also ascertained that pregnant women thought that MyPregn@ncy only provided added value when there were ‘problems’ in pregnancy, or when they were unsatisfied with their current communication with professionals. The role of the professional endorsing MyPregn@ncy turned out to be the most important factor in the decision to start using MyPregn@ncy. Previous professional adopters of the tool can therefore play an important role in the implementation process.

In comparing the data on uncomplicated births in the study region with the overall Dutch data [23,24,25], no trend toward improvement was detected. National data showed relatively small to no alteration in all of the investigated conditions over the past few years, ranging from −0.8% for vaginal non-instrumental delivery, to +0.1% for delivery at 37–42 weeks.

A recent meta-analysis found that participating in clinical trials improves outcomes in women’s health, irrespective of whether the intervention was effective or not [30]. This is often called the Hawthorne effect [31]. Our study suggested an increase in the number of uncomplicated births during the study period, although this was not statistically significant. We believe the trend toward better outcomes may be due to health care professionals being informed about the study, even before the start of their active participation or their offer of the intervention. By simply performing this study, the involved maternity health care centers and practitioners developed a higher awareness of women’s participation in their care, independent of the clusters and start of the intervention. Furthermore, the involved region initiated multiple quality developments in pregnancy and childbirth during the study. These efforts presumably had an additional positive contribution toward the improvements seen in uncomplicated birth rates.

Women in the control group, as well as in the intervention group, perceived the quality of maternity care received as very high. This implies that maternity care was already of relatively high quality. As the domains ‘autonomy’ and ‘choice and continuity’ were rated proportionally worse, these two issues constitute problems to be addressed in the future.

Health care professionals rated the quality of collaboration in the maternity care field as comparable between before and after the introduction of MyPregn@ncy, with the exception of hospital-based midwives. This finding indicates that the region-wide offer and use of an innovative PHR might endorse hospital-based midwives to have a more visible position in the regional maternity care network.

The main strength of this study is the high number of pregnant women included and the participation of all maternity health care professionals in the study region. Due to the engagement of a wide variety of health care professionals, it can be considered a widely supported and multidisciplinary study.

We acknowledge several shortcomings in the present study, including the small number of pregnant women actively using MyPregn@ncy and the absence of the active involvement of pregnant women in designing the implementation elements. As stated in our previously published paper concerning the process evaluation of the study, such user involvement in the design of a PHR contributes to its successful implementation and integration in standard care [14].

Given current national developments, the use of PHRs in health care will likely continue to grow in the coming years. While an introduction and cautious development of PHRs have been seen in the last ten years, the next ten years are expected to bring their faster and more decisive realization to many more people in health care. We recognize that the Dutch government is making a concerted effort to facilitate and stimulate the availability and usage of PHRs to all people. The support of the Dutch government is necessary to realize the potential of PHRs. With these innovations—through which each individual can access, manage, and share their health information in a private, secure, and confidential online environment—we empower patients, facilitate communication between health care professionals in the patients’ network, and thus improve health outcomes.

## 5. Conclusions

We conclude that the offer of the innovative intervention MyPregn@ncy had no significant effect on the percentage of uncomplicated births, the primary outcome of this study. The raw data suggested a positive effect, but this was not significant after correction for time. This positive pattern was caused by the rising trend in uncomplicated birth rates in the total eligible population during the study period. This local trend, which was less visible in the national database, may have been related to subtle changes toward an improved collaborative culture among the local professionals who participated in the study. Taking this into consideration, we posit that person-centered collaboration rewards, regardless of the intervention itself.

## Figures and Tables

**Figure 1 ijerph-18-10343-f001:**
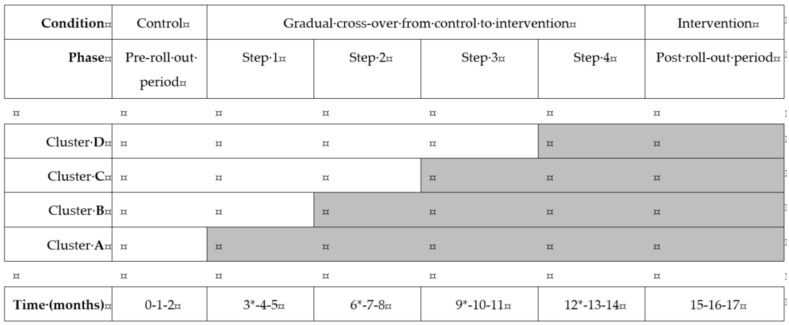
Schematic overview of the stepped-wedge study design. All clusters started in the control condition in the pre-roll out period (3 months). Clusters A-B-C-D then gradually crossed-over to the intervention condition after 3, 6, 9 and 12 months, respectively (*). Women were followed throughout pregnancy until birth. Outcome data collection continued until 6 months after the last inclusion during the post roll-out period.

**Figure 2 ijerph-18-10343-f002:**
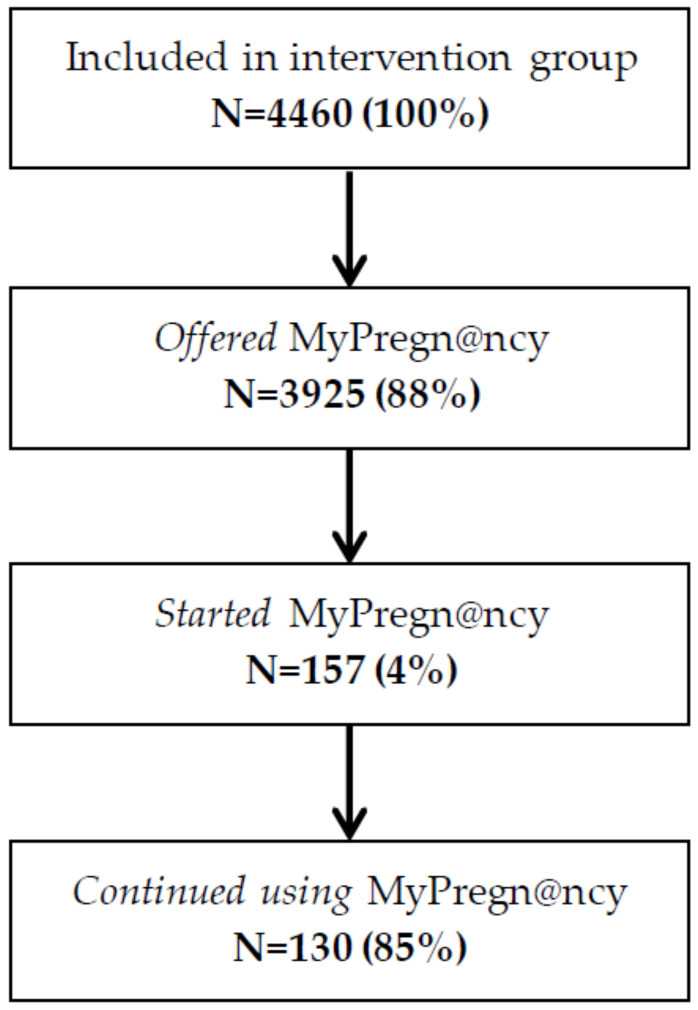
Uptake of the MyPregn@ncy intervention.

**Figure 3 ijerph-18-10343-f003:**
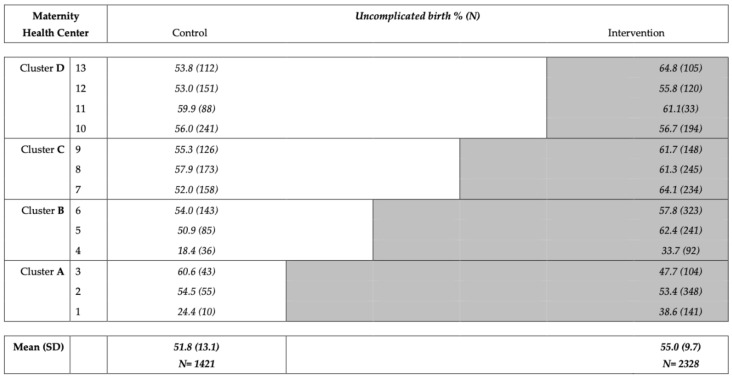
Percentage and numbers of uncomplicated births per maternity health center and overall (total 95% confidence interval) before and after the introduction of My Pregn@ncy.

**Figure 4 ijerph-18-10343-f004:**
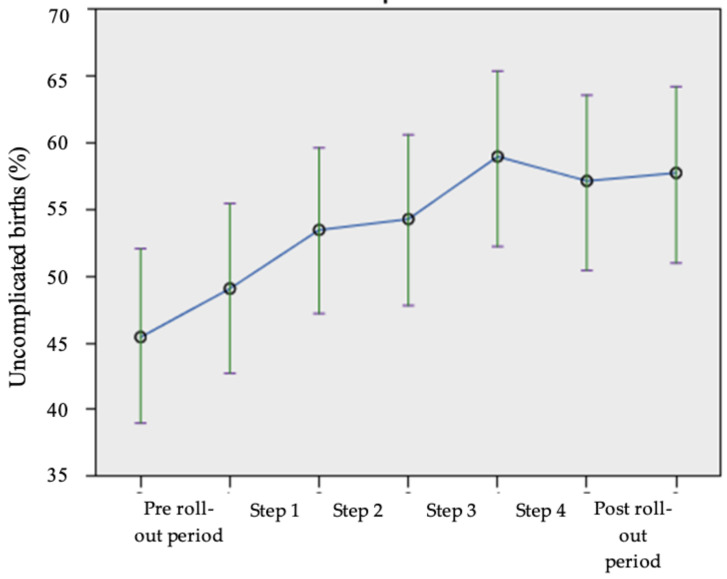
Total percentage (control and intervention group) uncomplicated births in each step of the study. Error bars represent 95% confidence intervals.

**Figure 5 ijerph-18-10343-f005:**
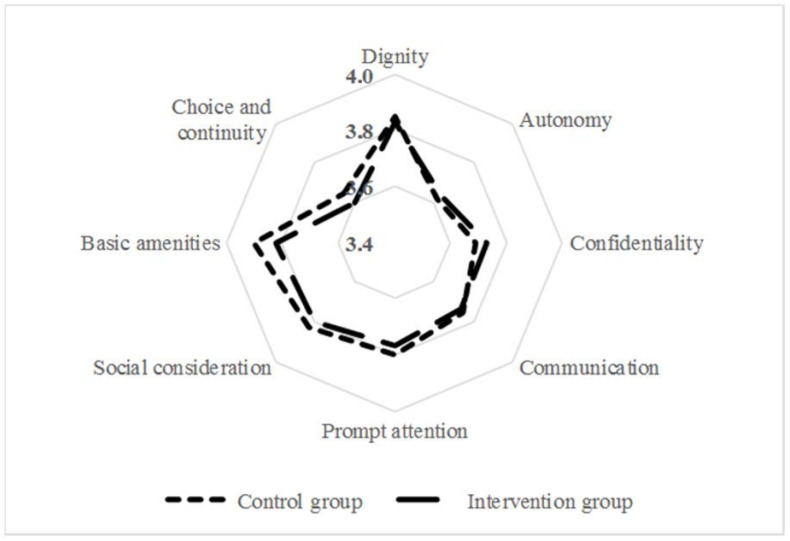
Women’s mean perspectives on received maternal care in the control and intervention groups.

**Table 1 ijerph-18-10343-t001:** Characteristics of the study participants (N = 7350).

	Control Group N = 2890	Intervention Group N = 4460
	N (%)	N (%)
Age (years)		
<20	17 (0.6)	36 (0.8)
20–24	215 (7.4)	308 (6.9)
25–29	818 (28.3)	1271 (28.5)
30–34	1193 (41.3)	1882 (42.2)
35–39	545 (18.9)	816 (18.3)
≥40	102 (3.5)	147 (3.3)
Ethnicity		
Dutch	2433 (84.2)	3702 (83.0)
Other	457 (15.8)	758 (17.0)
Parity		
Primary	1357 (47.0)	2203 (49.4)
Second	1098 (38.0)	1570 (35.2)
Third of more	435 (15.0)	687 (15.4)

**Table 2 ijerph-18-10343-t002:** Uncomplicated births.

	Control Group N = 2890	Intervention Group N = 4460	
	N (%); CI *(%)	N (%)	*p*-value
Complete data	2746 (95.0)	4233 (94.9)	
Uncomplicated births	1421 (51.8); 50.1–53.9	2328 (55.0); 53.5–56.5	0.289

* CI: 95% confidence interval.

**Table 3 ijerph-18-10343-t003:** Collaboration between health professionals.

Profession	Before MyPregn@ncy		After MyPregn@ncy		
	N	Score	N	Score	*p*-value ***
Community-based midwives	35	7.0	37	7.0	0.45
Hospital-based midwives	15	6.3	17	7.2	0.04
Obstetricians (in training)	22	6.9	24	6.9	0.94
Pediatricians	10	7.0	8	7.0	1.00
Maternity care assistants	5	6.8	5	7.8	0.32
Youth health nurses	29	6.9	17	6.9	0.94
Youth health doctors	13	6.8	4	6.5	0.50
Total (N; mean ± SD)	129	6.7 ± 0.2	112	6.9 ± 0.2	0.29

* Based on independent T-tests per profession group; overall results based on linear mixed model testin.

## Data Availability

The data presented in this study are available on request from the corresponding author.

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
