# Peer review of "Effects of a Personal Health Record in Maternity Care: A Stepped-Wedge Trial"

_ijerph, 2021, doi:10.3390/ijerph181910343_

Round 1

Reviewer 1 Report

1 Introduction

The introduction provides only an insufficient insight into the current state of research on PHRs. What evidence is available on the use of PHRs to improve health care? Which forms of PHR can be distinguished and what is the impact on outcomes? In addition, it would be good if a few key data on maternal health and health care in the Netherlands were added. Also with regard to where care problems exist.
What is the general situation regarding the use of PHRs in the Netherlands and the willingness to use digital health applications?

2 Material and Methods
2.1 Setting
Maternity health care (lines 58 -60) should be presented in a more differentiated way (see also Introduction). What is the usual care process? The Dutch Perinatal Registry should be mentioned here. The professional groups involved in maternity health care should also be briefly specified. 

2.2 Intervention
The description should be sharpened, as it is not easy to imagine how women access the website and how they enter and process the information. Perhaps an illustration with a screenshot would be helpful.
Which team "all team members" (line 80) is meant here? How was informed consent obtained for the use of the website and the data? Clarify the reference (lines 93-94).

2.3 Implementation strategy
What was the training of the health care providers like? Was it compulsory or voluntary?
How can one imagine the invitation by the health care provider (line 92)? Was the website presented to the women in the practice or was this done individually by the women themselves?

2.4 Study design
Were there any specifications for the inclusion of the women in the study? How were the different stages of pregnancy dealt with? Only a general reference is found here in line 121.
What differences exist between the maternal healthcare facilities with regard to the care of pregnant women and what influence does this have on the outcomes considered?
The allocation of the maternal care facilities to the clusters, especially the maternity health care centres, is not clearly comprehensible. This information could be added to Figure 1.
There is no information on how the data from the different surveys are linked, e.g. use of the PHR and the usercodes from the Dutch Perinal Registry or for the survey.
Why were only 50% of the participants invited? 

2.5 Outcome measure
Primary outcome: The definition should be backed up with relevant literature. There is only a very general reference found here in lines 130-131.
What are the underlying hypotheses/assumptions so that it can be expected that a change in the primary outcome is likely?

3. Results
How is the differentiation between control and intervention group achieved? How was the control group formed? What is the difference between the control group and the intervention group?
The proportion of women who started the MyPregn@ncy is very small. Here it would be important to know key data on these women (see Table 1). What exactly was the distribution between the maternity care facilities; it seems as if participation in the MyPregn@ncy took place primarily in one centre.
Is it even possible to draw conclusions about the primary outcome based on the participation rate in MyPregn@ncy?
Were the women who used MyPregn@ncy also interviewed? How do the results compare to the other women in the control and intervention groups?

4. Discussion
The following aspects should still be considered: 
- What was the reason for the difficulty in recruiting women to use MyPregn@ncy? Here, findings from the process evaluation could be useful. What recommendations result from this?
- Which topics should be integrated into MyPregn@ncy so that it is more attractive to women?
- Which care processes should MyPregn@ncy address so that an add value can be achieved in maternal healthcare? How should the structures be designed so that this can be achieved?

Author Response

Dear reviewer,

We thank you for the reviewers’ report and the opportunity to send a revised manuscript ‘Effects of a Personal Health Record in maternity care: a stepped-wedge trial’. We truly appreciate your suggestions for improvement of the manuscript. A point-by-point response to the reviewers’ comments is described below and accompanying changes in the revised manuscript are all tracked. We believe the manuscript has been improved thanks to the reviewers’ report and we hope it qualifies to be published.

Reviewer #1

1 Introduction

The introduction provides only an insufficient insight into the current state of research on PHRs. What evidence is available on the use of PHRs to improve health care? Which forms of PHR can be distinguished and what is the impact on outcomes? In addition, it would be good if a few key data on maternal health and health care in the Netherlands were added. Also, with regard to where care problems exist.
What is the general situation regarding the use of PHRs in the Netherlands and the willingness to use digital health applications
?

We agree with the reviewer and added more information in the Introduction section that answers the questions mentioned.

2 Material and Methods
2.1 Setting
Maternity health care (lines 58 -60) should be presented in a more differentiated way (see also Introduction). What is the usual care process? The Dutch Perinatal Registry should be mentioned here. The professional groups involved in maternity health care should also be briefly specified. 

We thank the reviewer for these critical suggestions, which we agree on. We have added the maternity health care in a more differentiated way, including the different involved professionals and we mentioned the Dutch Perinatal Registry here.

2.2 Intervention
The description should be sharpened, as it is not easy to imagine how women access the website and how they enter and process the information. Perhaps an illustration with a screenshot would be helpful.
Which team "all team members" (line 80) is meant here? How was informed consent obtained for the use of the website and the data? Clarify the reference (lines 93-94).

We understand this feedback and we have explained more about the access to the website and all team members in the text. We added how informed consent was obtained.

2.3 Implementation strategy
What was the training of the health care providers like? Was it compulsory or voluntary?
How can one imagine the invitation by the health care provider (line 92)? Was the website presented to the women in the practice or was this done individually by the women themselves?

We thank the reviewer for this feedback, and we have explained the answers to these questions in the text.

2.4 Study design
Were there any specifications for the inclusion of the women in the study? How were the different stages of pregnancy dealt with? Only a general reference is found here in line 121.
What differences exist between the maternal healthcare facilities with regard to the care of pregnant women and what influence does this have on the outcomes considered?
The allocation of the maternal care facilities to the clusters, especially the maternity health care centres, is not clearly comprehensible. This information could be added to Figure 1.
There is no information on how the data from the different surveys are linked, e.g. use of the PHR and the usercodes from the Dutch Perinal Registry or for the survey.
Why were only 50% of the participants invited? 

The intervention in this study is the introduction of MyPregn@ncy by offering to pregnant women to use this tool. All pregnant women in the maternity health centers in the intervention condition were offered to use MyPregn@ncy. We analysis the results at group level. Figure 2 shows that 88% off all women was offered MyPregn@ncy in the maternity health centers. We chose to include all different maternity health centers in the area including primary, secondary and tertiary care. In this way it facilitated multidisciplinary collaboration between health care professionals. Thereby, this is a representative representation of the Netherlands. We added this information in the article. In the ‘Study design section’ we describe that cluster A and B allocated each one of the hospitals. We invited 50% participants on the rational reason. Each step of 3 months was divided in six weeks of preparation and inclusion of the women and six weeks data collection. This explanation is added in the text.

2.5 Outcome measure
Primary outcome: The definition should be backed up with relevant literature. There is only a very general reference found here in lines 130-131.
What are the underlying hypotheses/assumptions so that it can be expected that a change in the primary outcome is likely?

We thank the reviewer for this point and realize that references for the definition improves the manuscript.

  1. Results
    How is the differentiation between control and intervention group achieved? How was the control group formed? What is the difference between the control group and the intervention group?
    The proportion of women who started the MyPregn@ncy is very small. Here it would be important to know key data on these women (see Table 1). What exactly was the distribution between the maternity care facilities; it seems as if participation in the MyPregn@ncy took place primarily in one centre.
    Is it even possible to draw conclusions about the primary outcome based on the participation rate in MyPregn@ncy?
    Were the women who used MyPregn@ncy also interviewed? How do the results compare to the other women in the control and intervention groups?

We randomized on cluster level, each cluster containing three of four maternity health centers with an intention-to-treat principle. We describe the characteristics of the study participants in table 1. Because we analyzed on groups level, we don’t have extra information of the women who used MyPregn@ncy. We understand the point of criticism if it is possible to draw conclusions based on the participation rate. Therefore, we started the Conclusion Section with: ‘Although, the participation rate was low’

  1. Discussion
    The following aspects should still be considered: 
    - What was the reason for the difficulty in recruiting women to use MyPregn@ncy? Here, findings from the process evaluation could be useful. What recommendations result from this?
    - Which topics should be integrated into MyPregn@ncy so that it is more attractive to women?
    - Which care processes should MyPregn@ncy address so that an add value can be achieved in maternal healthcare? How should the structures be designed so that this can be achieved?

The questions in this feedback are answered in our article already published ‘Personal Health Records in Maternity care ‘a process evaluation’ (14). We agree with the reviewer that a summary of this, more extensively, also should be included in this discussion. We added this in the ‘Discussion’ section.

As required by the reviewers we checked the article for spell checks and made tracked some spell improvements in the text. 

Reviewer 2 Report

This article evaluated the effect of an electric personal health record for pregnant women on uncomplicated births and related outcomes in the Netherlands. The manuscript describes an important finding of the effect of a personal health record in maternal, newborn, and child health. It is worth being disseminated after possible modifications based on the following comments, if they are relevant.

  1. It is recommended to fill out the CONSORT checklist extended for stepped-wedge cluster randomized trials (Table 3 of the following article: https://www.bmj.com/content/363/bmj.k1614) and attach it to the manuscript as a supporting information in order to make sure that all the necessary reporting items are included in the manuscript.
  2. The description of the main result is not clear in the Abstract and the main text. Was it statistically significant? It is informative to report the 95% confidence interval (1-53.9% vs. 53.5-56.5%) and the result of the significant testing (p=0.289) for the difference in the uncomplicated birth rate between the group, although the description “Estimated means showed that detected differences were due to time instead of the intervention.” implies that the difference was not statistically significant.
  3. Related to the comment above, overlapped confidence interval (1-53.9% vs. 53.5-56.5%) is small in the difference of uncomplicated birth rate between the groups. Although this result accounted for cluster effects, I still wonder why the p-value is so large (p=0.289) under a relatively large sample size. I would recommend the authors to check the results once again. According to a quick calculation on the independence between the groups and uncomplicated births (without considering clustering), p=0.008 in the chi-square test (of course, this result is not precise, though). If the difference was not statistically significant, the authors might want to describe possible reasons behind the differences in the size of clusters and those in the percentages of women who used MyPregn@ncy and had uncomplicated birth.
  4. The introduction section is extremely short. The authors may want to add a paragraph to introduce how personal health records were used in maternal, newborn, and child health. For example, WHO Recommendations on home-based records for maternal, newborn, and child health (https://apps.who.int/iris/bitstream/handle/10665/274277/9789241550352-eng.pdf) and articles reviewed in the recommendations may be relevant.
  5. Related to the comments above, the authors may want to consider addressing why personal health records like MyPregn@ncy is required in health facilities that had already provided a relatively high-quality care, by citing the relevant literature. What kinds of gaps did the authors identify before the introduction of the app and how did the authors conclude the app like MyPregn@ncy could address the gaps?
  6. Related to the comment above, the reference list is short (17 articles were cited).
  7. How was the minimum sample size calculated and set “a random sample of 50%” (Line 146) to achieve the minimum sample size? The authors may want to clarify that to show that the analysis has a sufficient statistical power.
  8. What was the method of a random selection of participants? Was it a systematic random sampling under which healthcare professionals invited women once every two of them? Please clarify.
  9. In this study, the number of study participants were 7350 (n=4,460 in the intervention group and n=2,890 in the control group). It seems that 7,350 is the number of women who answered the questionnaire at their first postnatal visit. (Lines 146-). If the authors kept records, they may want to report the number of women who received an information about the study (Lines 91-), the number of women who agreed to participate in the study, the number of women who instructed about MyPregn@ncy, and the number of women who were invited to answer the questionnaire, in addition to the number of women who answered the questionnaire. The authors may want to provide the chart of participant selection for both groups, as is recommended by CONSORT Statement, by including these data.
  10. It is unclear the data collection methods for ReproQ questionnaire and other questionnaire items, if any. Is it a self-administered questionnaire or interview questionnaire?
  11. It seems that the study participants of this study are both pregnant women and health care professionals in this study. However, in the Methods section, eligibility criteria, recruitment, and data collection methods for health care professionals were summarized in 2.5 Outcome measures. The authors may want to restructure each subsection in the Methods section to explain the necessary reporting items both of the pregnant women survey and the health care professional survey.
  12. What language was used in the questionnaire? This information is important for researchers who consider using the scales used in this study. Explain that the scales used in this study was used in the language used in this study, if it is correct.
  13. The statistical analysis model seems to be appropriate. However, it might not be clear how “fixed effects for intervention and step, and a random effect for health care center” in the model. It may be useful to add a reference to the following article: https://doi.org/10.1016/j.cct.2006.05.007 for readers who are interested in understanding the model.
  14. Were the covariates included (as the fixed effect) for the regression model? Under the possible confounding by time trend, some key articles related to the stepped widget cluster randomized trial recommend to adjust for potential confounders in addition to time trend (for example: https://www.bmj.com/content/350/bmj.h391). The authors may need to justify why the model did not include potential confounders if it did not; otherwise, it should be clarified in the Methods section that these were adjusted for.
  15. The estimation may produce wider variances in a mixed-effect regression when the sizes of clusters differ substantially. For readers to understand cluster heterogeneity, it would be beneficial to present the number of observations by health center and step. The authors might want to consider presenting the number of women in all combinations of health center and step, in addition to the percentage of uncomplicated births, in Figure 3.
  16. Is Figure 4 based on the intervention group only or pooled data on both groups? Please clarify in the footnote or title of the figure.

Author Response

Dear reviewer,

We thank you for the reviewers’ report and the opportunity to send a revised manuscript ‘Effects of a Personal Health Record in maternity care: a stepped-wedge trial’. We truly appreciate your suggestions for improvement of the manuscript. A point-by-point response to the reviewers’ comments is described below and accompanying changes in the revised manuscript are all tracked. We believe the manuscript has been improved thanks to the reviewers’ report and we hope it qualifies to be published.

  1. It is recommended to fill out the CONSORT checklist extended for stepped-wedge cluster randomized trials (Table 3 of the following article: https://www.bmj.com/content/363/bmj.k1614) and attach it to the manuscript as a supporting information in order to make sure that all the necessary reporting items are included in the manuscript.

We thank the reviewer for this reference. We used the articles of Karla Hemming for our study design. We used the checklist in the revised article.

  1. The description of the main result is not clear in the Abstract and the main text. Was it statistically significant? It is informative to report the 95% confidence interval (1-53.9% vs. 53.5-56.5%) and the result of the significant testing (p=0.289) for the difference in the uncomplicated birth rate between the group, although the description “Estimated means showed that detected differences were due to time instead of the intervention.” implies that the difference was not statistically significant.

We thank the reviewer for this question. The reviewer is right, the differences in the stepped-wedge design were due to time instead of the intervention. To clarify this, we added at the P value at text at figure 4.

We fully agree that it is informative to report the confidence interval and the results of the significant testing in the Abstract. We have added this information.

  1. Related to the comment above, overlapped confidence interval (1-53.9% vs. 53.5-56.5%) is small in the difference of uncomplicated birth rate between the groups. Although this result accounted for cluster effects, I still wonder why the p-value is so large (p=0.289) under a relatively large sample size. I would recommend the authors to check the results once again. According to a quick calculation on the independence between the groups and uncomplicated births (without considering clustering), p=0.008 in the chi-square test (of course, this result is not precise, though). If the difference was not statistically significant, the authors might want to describe possible reasons behind the differences in the size of clusters and those in the percentages of women who used MyPregn@ncy and had uncomplicated birth.

We thank the reviewer for this accurate point. As recommended, we have checked this data once again and the data are correct. We agree that because of the small difference in both groups, the expectation is a lower p value. However, because of the stepped-wedge analysis with including the covariates as the fixed effect, the estimated means showed that the differences in uncomplicated births were due to time instead of the intervention.

  1. The introduction section is extremely short. The authors may want to add a paragraph to introduce how personal health records were used in maternal, newborn, and child health. For example, WHO Recommendations on home-based records for maternal, newborn, and child health (https://apps.who.int/iris/bitstream/handle/10665/274277/9789241550352-eng.pdf) and articles reviewed in the recommendations may be relevant.

We agree with the reviewer. The other reviewer had corresponding feedback. We thank the reviewer for his/her suggestion. We used the recommended article ‘Understanding women’s, caregivers’, and providers’ experiences with home-based records: A systematic review of qualitative studies.’ We have added more information in the Introduction section that answers the questions mentioned.

  1. Related to the comments above, the authors may want to consider addressing why personal health records like MyPregn@ncy is required in health facilities that had already provided a relatively high-quality care, by citing the relevant literature. What kinds of gaps did the authors identify before the introduction of the app and how did the authors conclude the app like MyPregn@ncy could address the gaps?

We also agree also with this comment, we have explained the ‘gap’ and why MyPregn@ncy could address the gap in the Introduction section.

  1. Related to the comment above, the reference list is short (17 articles were cited).

With the revision of this article and processing the comments, we added more information in this article and with this, more references. The total is 29 in the revised article.

  1. How was the minimum sample size calculated and set “a random sample of 50%” (Line 146) to achieve the minimum sample size? The authors may want to clarify that to show that the analysis has a sufficient statistical power.

For this study the sample size calculations were based on the primary outcome measure and described in the published Study protocol article. ‘Improving maternity care using a personal health record: study protocol for a stepped-wedge, randomised, controlled trial. (2016) DOI 10/1186/s13063-016-1326-0.

  1. What was the method of a random selection of participants? Was it a systematic random sampling under which healthcare professionals invited women once every two of them? Please clarify.

We thank the reviewer for this question. In this stepped-wedge design we randomized clusters of maternity health centers and all pregnant women in these centers as the study design describes. If a maternity center cross over to the intervention condition, all pregnant women in these centers were all informed about MyPregn@ncy. To clarify this more we have added ‘pregnant women in clusters of maternity health centers’ and ‘pregnant women in all maternity health centers. In this way we believe the cluster randomization is clearer.

  1. In this study, the number of study participants were 7350 (n=4,460 in the intervention group and n=2,890 in the control group). It seems that 7,350 is the number of women who answered the questionnaire at their first postnatal visit. (Lines 146-). If the authors kept records, they may want to report the number of women who received an information about the study (Lines 91-), the number of women who agreed to participate in the study, the number of women who instructed about MyPregn@ncy, and the number of women who were invited to answer the questionnaire, in addition to the number of women who answered the questionnaire. The authors may want to provide the chart of participant selection for both groups, as is recommended by CONSORT Statement, by including these data.

We thank the reviewer for this critical point. The intervention in this study was the introduction of MyPregn@ncy to individual pregnant women, independent of gestational age or care setting, i.e., she was offered the possibility to start MyPregn@ncy. Data analysis for the stepped-wedge design was performed according to an intention-to-treat protocol. We added this information in 2.2. Intervention.

  1. It is unclear the data collection methods for ReproQ questionnaire and other questionnaire items, if any. Is it a self-administered questionnaire or interview questionnaire?

We thank the reviewer or this fair point. It is self-administered, and of course we have added this in the text.

  1. It seems that the study participants of this study are both pregnant women and health care professionals in this study. However, in the Methods section, eligibility criteria, recruitment, and data collection methods for health care professionals were summarized in 2.5 Outcome measures. The authors may want to restructure each subsection in the Methods section to explain the necessary reporting items both of the pregnant women survey and the health care professional survey.

We agree with the reviewer that the data collection is part of 2.5 ‘Outcome measures’. We explained the necessary reporting items.

  1. What language was used in the questionnaire? This information is important for researchers who consider using the scales used in this study. Explain that the scales used in this study was used in the language used in this study, if it is correct.

We agree with the reviewer, and we explain the used language in the text.

  1. The statistical analysis model seems to be appropriate. However, it might not be clear how “fixed effects for intervention and step, and a random effect for health care center” in the model. It may be useful to add a reference to the following article: https://doi.org/10.1016/j.cct.2006.05.007 for readers who are interested in understanding the model.

We really appreciate this advice from the reviewer and fully agree this is useful. We have added this reference.

  1. Were the covariates included (as the fixed effect) for the regression model? Under the possible confounding by time trend, some key articles related to the stepped widget cluster randomized trial recommend to adjust for potential confounders in addition to time trend (for example: https://www.bmj.com/content/350/bmj.h391). The authors may need to justify why the model did not include potential confounders if it did not; otherwise, it should be clarified in the Methods section that these were adjusted for.

We believed we described this in 2.6 Statistical analysis:

‘To estimate the intervention effect on the primary outcome, a generalized linear mixed-model with logit link and binomial distribution was applied, with as dependent variable the uncomplicated birth (yes/no), fixed effects for intervention and step, and a random effect for health care center with a variance components covariance structure. Hence, the analysis had some similarities with a time-series analysis with multiple time points before and after the intervention.’

  1. The estimation may produce wider variances in a mixed-effect regression when the sizes of clusters differ substantially. For readers to understand cluster heterogeneity, it would be beneficial to present the number of observations by health center and step. The authors might want to consider presenting the number of women in all combinations of health center and step, in addition to the percentage of uncomplicated births, in Figure 3.

We agree with the reviewer that this would be beneficial and added the numbers in the new Figure 3.

  1. Is Figure 4 based on the intervention group only or pooled data on both groups? Please clarify in the footnote or title of the figure.

We agree with the reviewer. Figure 4 is based on both groups. We clarify this in the text of the figure.

Round 2

Reviewer 1 Report

Dear authors,
I am glad that the feedback was helpful for you and that you have already implemented many of the recommendations in your manuscript. The article has clearly gained in clarity, especially the additions to the care system in the Netherlands are very helpful.
From my point of view, however, there is still one sticking point in the article. This concerns firstly the division into the control and intervention group and secondly the statistical comparison between the control and intervention group.
On the first point: What is the composition of the control group? I explained it to myself in such a way that it is the women who were not included in step 1 to step 3. It would be helpful if you could describe this in detail in the text and, for example, add the respective numbers of cases for the control and intervention groups in Figure 1.
The second point is somewhat more complex: If I have understood correctly, only a total of 157 participated in the intervention, i.e. used MyPreg@ncy. There was no other intervention offered to the women. Is that correct? If that is correct, then you would actually have only one genuine intervention group of N=157 and the rest of the women in the intervention group you have presented have only been given the opportunity to participate in the intervention. As a result, they are not really different from the control group. I.e. you could then only make a comparison between the control group and the 157 women (the genuine intervention group). If the women in the intervention group you defined have received further measures, then I would recommend a comparison between control group - intervention group without MyPreg@ncy and intervention group with MyPreg@ncy.

Author Response

Dear Reviewer,

We thank you for this feedback. Our response to the two points in the reviewer’s comments are described below and accompanying changes in the revised manuscript are all tracked.

On the first point: What is the composition of the control group? I explained it to myself in such a way that it is the women who were not included in step 1 to step 3. It would be helpful if you could describe this in detail in the text and, for example, add the respective numbers of cases for the control and intervention groups in Figure 1.

We thank the reviewer for this point, the explanation mentioned is right. We understand that it is important to explain very clearly which pregnant women in which maternity health centers were included in the control condition and which pregnant women in which maternity health center were included in the intervention condition. To explain this more clearly we added in de section ‘study design and cluster randomization’ in the text:

Figure 1 illustrates the study design, including a pre roll out period, four cross-over-points, and a post roll out period. The intervention group in this stepped-wedge trial is composed of all pregnant women of the maternity health centers in cluster A in step 1, all pregnant women of the maternity health centers in cluster A and B in step 2, all pregnant women of all maternity centers in clusters A, B and C in step 3, and all pregnant women in all the maternity centers in cluster A, B, C and D in step 4 and the post roll out period. The control group was composed out of all other women, including all pregnant women of the maternity centers in cluster A, B, C and D in the pre roll out period, all pregnant women of the maternity centers in cluster B, C and D in step 1, all pregnant women of the maternity centers in the clusters C and D in step 2 and all pregnant women of the maternity centers in cluster D in step 3.

At the start of the study, all pregnant women in maternity health centers started in the control condition; at the end of the study, all pregnant women in maternity centers had switched to the intervention condition.

The second point is somewhat more complex: If I have understood correctly, only a total of 157 participated in the intervention, i.e. used MyPreg@ncy. There was no other intervention offered to the women. Is that correct? If that is correct, then you would actually have only one genuine intervention group of N=157 and the rest of the women in the intervention group you have presented have only been given the opportunity to participate in the intervention. As a result, they are not really different from the control group. I.e. you could then only make a comparison between the control group and the 157 women (the genuine intervention group). If the women in the intervention group you defined have received further measures, then I would recommend a comparison between control group - intervention group without MyPreg@ncy and intervention group with MyPreg@ncy.

For this point, the reviewer did not understand this correctly. As described in the statistic section, we intended an intention-to-treat analysis with the intervention ‘offering MyPregn@ncy’ instead of the use in the end of MyPregn@ncy. Therefore, all women in the intervention period (N=4460) are in the intervention group as we also describe in the results section. To also provide insights in the use of MyPregn@ncy inside the intervention group we showed the uptake and use of MyPregn@ncy in figure 2. But all results are based on the intention-to-treat analysis and the corresponding numbers mentioned in table 1; the control group of 2890 pregnant women and the intervention group of 4460 pregnant women. To make this more clear, we made changes in the manuscript to explicitly make clear that the intervention is offering MyPregn@ncy and not the use in the end. In this way we believe it will be clearer for the readers.

We hope that this revision is well received by the reviewer and will meet the standards of the International Journal of Environment Research and Public Health and will be considered for publication.

Kind regards,

Carola Groenen MSc., on behalf of all authors.
